# Heterogeneity in the association between youth unemployment and mental health later in life: a quantile regression analysis of longitudinal data from English schoolchildren

Liam Wright  , Jenny Head, Stephen Jivraj

► Prepublication history and additional online supplemental material for this paper are available online. To view these files, please visit the journal online. To view these files, please visit the journal online (http://dx.doi.org/10.1136/bmjopen-2020-047997).

Department of Epidemiology and Public Health, University College London, London, UK

**Correspondence to**
Liam Wright;
liam.wright.17@ucl.ac.uk

## ABSTRACT

**Objectives** An association between youth unemployment and poorer mental health later in life has been found in several countries. Little is known about whether this association is consistent across individuals or differs in strength. We adopt a quantile regression approach to explore heterogeneity in the association between youth unemployment and later mental health along the mental health distribution.

**Design** Prospective longitudinal cohort of secondary schoolchildren in England followed from age 13/14 in 2004 to age 25 in 2015.

**Setting** England, UK.

**Participants** 7707 participants interviewed at age 25.

**Primary and secondary outcome measures** 12-Item General Health Questionnaire (GHQ) Likert score, a measure of minor psychiatric morbidity.

**Results** Youth unemployment was related to worse mental health at age 25. The association was several times stronger at deciles of GHQ representing the poorest levels of mental health. This association was only partly attenuated when adjusting for confounding variables and for current employment status. In fully adjusted models not including current employment status, marginal effects at the 50th percentile were 0.73 (95% CI −0.05 to 1.54, b=0.11) points, while marginal effects at the 90th percentile were 3.76 (95% CI 1.82 to 5.83; b=0.58) points. The results were robust to different combinations of control variables.

**Conclusions** There is heterogeneity in the longitudinal association between youth unemployment and mental health, with associations more pronounced at higher levels of psychological ill health. Youth unemployment may signal clinically relevant future psychological problems among some individuals.

## INTRODUCTION

The pandemic of COVID-19 looks set to plunge the world into recession and unemployment rates have begun to rise sharply across the globe.[1] A large body of research finds that unemployed individuals have worse mental health and well-being than their peers,

**Strengths and limitations of this study**

► The study includes longitudinal data from a large sample of young English adults, enabling us to control for several background factors that may explain selection into unemployment.

► To our knowledge, this is the first study on the long-term association between youth unemployment and later mental health using data from adults who entered adulthood during the aftermath of the 2008 Global Financial Crisis.

► We use quantile regression to study whether the association is heterogeneous or consistent across all individuals.

► We run thousands of models to test whether our results are robust to alternative model specifications.

► While we control for several adolescent factors, we use observational data so results may not indicate causality.

with evidence from longitudinal and quasi-experimental studies suggesting the association between unemployment and mental health is causal.[2–4] Several theories have been proposed for why unemployment may negatively affect mental health,[5] including loss of material and non-material ('latent') benefits of employment.[6] Given this literature, there is concern that the increasing unemployment rates will lead to higher rates of mental ill health.[7]

The effect of unemployment on mental health may not be transient. Several studies suggest that unemployment 'scars', defined as an effect of (prior) unemployment and poor mental health persisting even on re-employment.[8] A focus of the literature has been on the long-term effect of youth unemployment specifically, with studies from the USA,[9] the UK,[10] Canada[11] and Sweden[12 13] finding associations between early unemployment and

subjective well-being, depression and anxiety symptoms measured decades later. These associations remain (where tested) after adjusting for adolescent mental health. This suggests that associations cannot be explained by health-related selection into unemployment.

Focusing on youth unemployment (typically defined as unemployment before age 25) is valuable for two reasons. First, unemployment rates are higher among young people and recessions have a disproportionate impact on the job prospects of younger workers.[10] Second, adolescence and young adulthood are important from a life course perspective—unemployment at these ages may have larger long-term effects. Unemployment early in the transition to adulthood has been argued to hinder maturation into normative adult roles and identity[12] and to set off 'chains of risk',[14] with initial disadvantage begetting further disadvantage, such as increased future unemployment risk and lower lifetime wages[15 16]—factors which are themselves related to poorer mental health. Adolescence and early adulthood are also sensitive periods of physiological and neurobehavioural development.[14 17] Stressors experienced during this age may have long term effects on physiology and stress responses, negatively impacting lifelong physical and mental health.

While an overall association between youth unemployment and mental health later in life appears well established, little empirical research has explored the reasons behind such a link. Further, little research has studied whether scarring is uniform across all formerly unemployed individuals or whether effects differ in strength or are confined to particular individuals. Yet, this information is useful for directing resources towards the most vulnerable and for providing insight into the life-course processes that may generate long-term effects.

In this paper, we extend the literature by using quantile regression to investigate heterogeneity in the strength of these long-term effects. Unlike in typical ordinary least squares regression, which estimates changes in the conditional mean of a dependent variable, quantile regression estimates changes at specified quantiles of the distribution (ie, 10th percentile, median, etc).[18] Quantile regression thus allows researchers to model change in both the location and shape of the distribution of a dependent variable in response to changes in other variables, and, accordingly, to see whether associations differ in direction or strength across the distribution.

A small number of studies have used quantile regression to investigate heterogeneity in the contemporary association between unemployment and mental health using data from working-age adults.[19–21] Only one study has looked at long-term effects but has looked at life satisfaction, specifically.[22] Each of these studies finds that the negative association between unemployment and mental health is stronger at poorest levels of mental health or well-being. This is consistent with studies of trajectories in mental well-being which show that responses to episodes of unemployment are not uniform, with a minority of individuals appearing more strongly affected.[23 24]

We hypothesise that long-term effects will be heterogeneous and that they will be strongest at quantiles representing the poorest levels of mental health. We reason that the strength of the proposed pathways linking youth unemployment to later mental health are likely to differ across individuals. For instance, the consequences of early unemployment for later labour market outcomes (ie, 'chains of risk') is unlikely to be uniform. A recent German study finds that the scarring effect of youth unemployment on future unemployment is primarily driven by a minority of formerly unemployed individuals experiencing particularly long periods of later unemployment,[16] while the literature on 'compensatory disadvantage' shows that some social adversities can be overcome by individuals with sufficient socioeconomic resources.[25] Further, studies of psychological resilience show that many individuals experience little psychological injury in response to many life stressors[26]—job loss included[24 27]—while others experience larger and longer-lasting effects. This literature also provides some evidence that those with pre-existing mental health problems are more negatively affected by adverse events.[28]

In this paper, we also test whether associations differ by gender. Previous research shows that the contemporary effect of unemployment is typically greater among males,[2] though studies using more recent data in contexts of greater female labour force participation do not find differences by gender.[29] We also test whether long-term associations remain after adjusting for current economic activity, in line with original conceptions of scarring as a long-term effect that is independent of later employment status.[8]

## METHODS
### Sample
We use data from Next Steps (formerly, the Longitudinal Study of Young People in England), a cohort of English schoolchildren recruited at age 13/14 in 2003/2004.[30] The cohort were followed annually for 7 years from age 13/14 to age 19/20 and surveyed again at age 25 in 2014/15. Participants were interviewed at each wave, with their primary and secondary caregivers also interviewed for the first four waves.

Participants were recruited using a two-stage stratified sampling design with participants selected from a sample of state and independent schools. A total of 15 770 individuals were originally recruited with individuals from ethnic minority backgrounds and schools with a high proportion of students on free school meals over-sampled (response rate: 74%). A sample boost of 352 ethnic minority participants was also added in age 16/17. From 2004 to 2010, only previous responders were followed across waves and per wave drop-out varied between 8% and 14%. At the 2014/2015 wave, efforts were made to contact previous non-responders. Our sample is all individuals who participated at the age 25 survey (n=7707; 47.8% of total sample).

## Measures

### Mental health at age 25

We measure mental health at age 25 using the 12-Item General Health Questionnaire (GHQ-12). The GHQ was developed as a screening tool for minor non-psychotic psychiatric morbidity in the general population[31] and has been found to be a valid measure of mood and anxiety disorders.[32] Items relate to the appearance of new, distressing phenomena and reduced functioning due to psychiatric problems compared with usual experience. Each item is scored on a four-point scale. We use the summed Likert score as our outcome variable. Higher scores indicate worse mental health (range 0–36).

An issue with this measure is that by using usual experience as the reference condition, the GHQ may not adequately capture chronic problems or long-term effects of unemployment.[33] However, a recent validation study finds that the GHQ displays good sensitivity for detecting depression[32] and in online supplemental figure S1, we show visual evidence from the United Kingdom Household Longitudinal Study that GHQ scores are related to long-term depression.

### Youth unemployment

We measure youth unemployment using a binary indicator for whether the participant experienced 6+ months (continuous) unemployment between October 2008 and May 2010 (approximately ages 18–20). This period corresponds to the first 20 months following the summer holiday after the normative end date of further education for this cohort. It also overlaps with the beginning of the Global Financial Crisis, following which youth unemployment rates rose precipitously worldwide.[10] We derive the variable from activity history data which participants provided at each wave on main activities carried out since their previous interview. Activities were selected from a list, with unemployment appearing as 'unemployed' or 'unemployed and looking for work'.

We chose a 6+ month cut-off to focus on longer-term periods of youth unemployment, excluding short-term frictional episodes of unemployment that frequently appear during school holidays.[34] In sensitivity analyses, we repeat models using cut-offs of 3+, 9+ and 12+months, instead. Note, we focus on unemployment, specifically, rather than not in employment, education or training (NEET)—another measure of exclusion from the labour market—given the more heterogeneous set of experiences that NEET incorporates, such as full-time caring, 'gap years' and long-term disability. As Furlong[34] notes, the heterogeneity within NEET "means that both research and policy must begin by disaggregating so as to be able to identify the distinct characteristics and needs of the various subgroups'.

## Covariates

We add several control variables to partly account for non-random selection into unemployment. To account for mental health-related selection, we use scores from the 12-item GHQ at ages 14/15 and 16/17. We use the *Caseness* score at age 14/15 (1 if has experienced the symptom more than usual, 0 otherwise; range 0–12). Participants were able to respond 'don't know' to each item at this interview (we assume this reflects not experiencing the symptom). We use the Likert score at age 16/17 (range 0–36). The GHQ-12 has been shown to have acceptable validity in adolescent samples.[29 30]

To account for physical health-related selection, we control for self-rated health at ages 14/15 and 16/17 (categories: very good, fairly good, not very good, not good at all) and whether the participant had a disability at age 14/15 or 15/16 (categories: no disability; disability, but schooling unaffected; disabled and schooling affected).

We also adjust for gender and ethnicity (categories: white, mixed, Indian, Pakistani, Bangladeshi, Black African, Black Caribbean, other), and to account for differences in social background, we include variables for the participant's family social class (categories: higher, intermediate, routine/manual, long-term unemployed) and highest parental education (further education or higher vs secondary or lower), both measured when the participant was aged 13/14. We also include a measure of neighbourhood deprivation at age 14/15, the Index of Multiple Deprivation 2004 (IMD), which is produced by the UK Government and captures local area deprivation across seven dimensions (income, employment, health, education, barriers to housing and services, living environment and crime). We use quintiles of IMD, with higher quintiles indicating greater deprivation.[35]

To account for differences in human capital, we control for educational attainment at age 25 (six categories: National Vocational Qualification levels 1–5, no qualifications). (Education data are not publicly available for ages prior to this.) Social adjustment and non-cognitive skills are likely to cause both labour market outcomes and mental health. As proxies for some of these, we include variables for positive attitude to school (age 14/15; summed response to 12-item measure, range 0–48), risk behaviours (age 13/14; summed response to 8-item measure on antisocial behaviour, alcohol, smoking and drug use in previous 12 months, range 0–8), and bullying victimisation (number of waves reported being bullied in prior 12 months, age 13/14 – 15/16, range 0–3). We also include a measure of internal locus of control (beliefs about the extent to which the participant controls their life), produced from confirmatory factor analysis of six questionnaire item. Further detail on this measure is provided in the online supplemental information.

Current economic activity at age 25 was derived from a question on main activity (categories: employed, education, unemployed, inactive).

## Statistical analysis

We use quantile regression to estimate the association between youth unemployment and GHQ scores at age 25 at each decile of the outcome variable. We run unadjusted models, covariate adjusted models and covariate adjusted

models with current economic activity as a further control variable. We use bootstrapping to estimate confidence intervals (500 replications) and apply survey weights supplied with the data to account for differences in the likelihood of recruitment to, and exit from, the study. To account for item-missingness, we use multiple imputation (m=28, burn-in=10), adding the survey weights as linear terms in the imputation models to partly account for weighting in our final analysis.[36] We use data from all sample members interviewed at age 25 in the imputations, but discard participants with missing outcome data in the quantile regressions (final sample n=7363). We conduct multiple imputations separately for male and female participants.

A potential issue with our analysis is that some control variables may bias associations. For instance, education may be a mediator and the use of proxies for social adjustment and non-cognitive skills could open backdoor paths between youth unemployment and later mental health.[37] Therefore, as a sensitivity analysis, we run a random selection of 20 000 models drawn from all possible combinations of the covariates defined above. To minimise the computational cost, we use only a single imputed dataset for these regressions.

Another potential issue is that we have included few controls for adolescent family background and socioeconomic position, which may cause residual confounding. Therefore, as another sensitivity analysis, we repeat our main analysis including further adjustment for family financial difficulties ('Thinking of how your household is managing on your total household income at the moment, would you say it was…': managing quite well, able to save or spend on leisure; just getting by, unable to save if wanted to; getting into difficulties), number of household children (categorical: 1, 2, 3, 4, 5+) and household type (two parent household, single parent household). Each of these variables was measured at age 13/14 from parental reports. Finally, as noted above, as a final sensitivity analysis, we repeat the main analysis model (no adjustment for current status) using different cut-offs to define youth unemployment: 3+, 9+ and 12+months unemployment.

Multiple imputations[38] and data analysis were carried out in R V.3.6.3.[39] The code to replicate this analysis is available at https://osf.io/3gwap/. Next Steps data are available through the UK Data Service.[30]

### Patient and public involvement
The public were not involved in the design, or conduct, or reporting, or dissemination plans of our research.

## RESULTS
### Descriptive statistics
Descriptive statistics are displayed in table 1. Individuals with 6+ months unemployment between ages 18 and 20 differ from those with less unemployment experience along a number of dimensions, including mental and physical health and socioeconomic background. This is consistent with other studies showing that selection into youth unemployment is not random.[40 41] There is little difference according to GHQ scores at age 16/17, which may suggest measurement error in this variable.

### Quantile regression
The results of the quantile regressions are presented in table 2. Except at the lowest quantiles of GHQ scores (which indicate good mental health), youth unemployment is associated with worse mental health at age 25 (column A). Associations are only partly attenuated controlling for adolescent mental health (column B) or adolescent mental health and other background characteristics (column C). Consistent with our hypothesis, associations are stronger at quantiles representing greater levels of poor mental health. For example, the predicted GHQ score at the 50th percentile among those with 6+months unemployment is 0.73 (95% CI –0.05 to 1.54, b=0.11) points greater than at the 50th percentile among those with less than 6 months unemployment. At the 90th percentiles, the difference is 3.76 (95% CI 1.82 to 5.83; b=0.58) points. Further adjustment for current employment status partly attenuates these estimates (column D), but associations remain substantial at highest quantiles. The results for models in columns C and D are displayed graphically in figure 1. (Full regression results are displayed in online supplemental tables S1–S6)

Quantile regression of GHQ-12 Likert scores at age 25 on 6+ months unemployment experience between ages 18 and 20. (A) Bivariate association, (B) adjustment for GHQ-12 scores at ages 14/15 and 16/17, (C) Model B plus adjustment for disability and self-rated health, educational attainment, risk behaviours, attitude to school and bullying victimisation, IMD quintile, parental socioeconomic class and education, gender, locus of control and ethnicity, (D) Model C plus adjustment for current economic activity, (E) Model C estimated for males only and (F) Model C estimated for females only.

Predicted GHQ-12 scores according to youth unemployment experience derived from fully adjusted model (excluding current activity) in column C are displayed in figure 2. (Other covariates kept at the sample means or modes.) Over 30% of those with 6+months unemployment are predicted to have GHQ Likert scores above 15, 10% points more than those with less youth unemployment experience. For context, one recent Swedish validation study found a cut-off of 11/12 points provides adequate sensitivity and specificity for detecting depression.[32]

Figure 3 shows the results of the adjusted model not including current economic activity stratifying by gender. Stronger associations are again found at poorest levels of mental health, and effect sizes are similar in each gender. The results are also displayed in columns E and F of table 2 (male and female, respectively). Full regression results are displayed in online supplemental information.

**Table 1** Descriptive statistics

| | Unweighted observed data | | | Weighted imputed data | |
| | <6+ Mmonths | 6+ months | | <6+ months | 6+ months |
| Variable | Unemployment | Unemployment | % Missing | Unemployment | Unemployment |
|---|---|---|---|---|---|
| n | 6589 (91.76%) | 592 (8.24%) | | 6599.90 (88.03%) | 897.13 (11.97%) |
| GHQ-12 @ age 25 | 11.53 (6) | 13.42 (7.59) | 4.46% | 11.71 (6.19) | 13.62 (7.65)* |
| Gender | | | | | |
| Male | 2799 (42.48%) | 317 (53.55%) | 0% | 3183.39 (48.23%) | 538.54 (60.03%)* |
| Female | 3790 (57.52%) | 275 (46.45%) | | 3416.51 (51.77%) | 358.59 (39.97%) |
| IMD | 22.56 (16.76) | 30.42 (18.13) | 8.78% | 22.18 (16.18) | 30.08 (18.26)* |
| Locus of Control | 0.05 (0.97) | −0.51 (1.14) | 12.72% | −0.05 (1.01) | −0.64 (1.13)* |
| Current Economic Activity | | | | | |
| Employed | 5569 (85.19%) | 370 (62.93%) | 1.12% | 5499.25 (83.32%) | 521.59 (58.14%)* |
| Education | 307 (4.7%) | 11 (1.87%) | | 271.68 (4.12%) | 12.64 (1.41%) |
| Inactive | 402 (6.15%) | 92 (15.65%) | | 523.71 (7.94%) | 182.18 (20.31%) |
| Unemployed | 259 (3.96%) | 115 (19.56%) | | 305.25 (4.63%) | 180.73 (20.15%) |
| GHQ-12 @ age 14/15 | 1.74 (2.53) | 2.06 (2.76) | 13.16% | 1.76 (2.58) | 2.1 (2.72)* |
| GHQ-12 @ age 16/17 | 10.5 (5.92) | 10.29 (6.48) | 20.85% | 10.34 (5.98) | 10.3 (6.53) |
| Self-rated health @ age 14/15 | | | | | |
| Very good | 2576 (45.54%) | 166 (34.51%) | 15.22% | 2902.29 (43.97%) | 306.18 (34.13%)* |
| Fairly good | 2903 (51.32%) | 288 (59.88%) | | 3401.69 (51.54%) | 520.99 (58.07%) |
| Not very good | 150 (2.65%) | 21 (4.37%) | | 238.64 (3.62%) | 58.41 (6.51%) |
| Not good at all | 28 (0.49%) | 6 (1.25%) | | 57.27 (0.87%) | 11.55 (1.29%) |
| Self-rated health @ age 16/17 | | | | | |
| Very good | 2922 (51.97%) | 215 (43.26%) | 15.67% | 3315.12 (50.23%) | 384.90 (42.9%)* |
| Fairly good | 2308 (41.05%) | 246 (49.5%) | | 2736.67 (41.47%) | 440.17 (49.06%) |
| Not very good | 329 (5.85%) | 29 (5.84%) | | 440.10 (6.67%) | 57.85 (6.45%) |
| Not good at all | 63 (1.12%) | 7 (1.41%) | | 108.00 (1.64%) | 14.21 (1.58%) |
| Disabled | | | | | |
| No | 5619 (87.7%) | 466 (80.48%) | 2.76% | 5633.49 (85.36%) | 701.73 (78.22%)* |
| Yes, school not affected | 476 (7.43%) | 47 (8.12%) | | 556.52 (8.43%) | 74.86 (8.34%) |
| Yes, school affected | 312 (4.87%) | 66 (11.4%) | | 409.89 (6.21%) | 120.54 (13.44%) |
| Risk behaviours | 0.76 (1.31) | 1.18 (1.65) | 12.24% | 0.96 (1.5) | 1.47 (1.84)* |
| Attitude to school | 33.49 (7.19) | 30.1 (8.02) | 10.72% | 32.28 (7.72) | 28.33 (8.26)* |
| # Waves bullied, 1–3 | 1.34 (1.15) | 1.61 (1.15) | 15.47% | 1.47 (1.15) | 1.72 (1.13)* |
| Qualifications | | | | | |
| NVQ 5 | 1167 (17.71%) | 19 (3.21%) | 0% | 927.05 (14.05%) | 16.90 (1.88%)* |
| NVQ 4 | 1846 (28.02%) | 57 (9.63%) | | 1611.71 (24.42%) | 65.08 (7.25%) |
| NVQ 3 | 1355 (20.56%) | 72 (12.16%) | | 1147.44 (17.39%) | 70.78 (7.89%) |
| NVQ 2 | 1350 (20.49%) | 189 (31.93%) | | 1613.46 (24.45%) | 252.11 (28.1%) |
| NVQ 1 | 497 (7.54%) | 165 (27.87%) | | 815.63 (12.36%) | 338.13 (37.69%) |
| No/other qual | 374 (5.68%) | 90 (15.2%) | | 484.60 (7.34%) | 154.14 (17.18%) |
| Parental NS-SEC | | | | | |
| Higher | 2393 (41.27%) | 117 (22.72%) | 12.33% | 2503.64 (37.93%) | 169.41 (18.88%)* |
| Intermediate | 1208 (20.83%) | 87 (16.89%) | | 1393.86 (21.12%) | 127.20 (14.18%) |
| Routine | 1889 (32.58%) | 259 (50.29%) | | 2363.27 (35.81%) | 516.22 (57.54%) |
| Long-Term Unemployed | 308 (5.31%) | 52 (10.1%) | | 339.13 (5.14%) | 84.30 (9.4%) |

Continued

**Table 1** Continued

| Variable | Unweighted observed data | | % Missing | Weighted imputed data | |
|---|---|---|---|---|---|
| | **<6+ Mmonths**<br>**Unemployment** | **6+ months**<br>**Unemployment** | | **<6+ months**<br>**Unemployment** | **6+ months**<br>**Unemployment** |
| Parental education | | | | | |
| Degree | 1080 (19.44%) | 52 (10.28%) | 15.71% | 1098.52 (16.64%) | 67.68 (7.54%)* |
| Other Higher Education | 939 (16.9%) | 64 (12.65%) | | 1042.91 (15.8%) | 91.76 (10.23%) |
| A-level | 973 (17.51%) | 77 (15.22%) | | 1141.75 (17.3%) | 128.93 (14.37%) |
| GCSE A-C | 1381 (24.86%) | 128 (25.3%) | | 1844.99 (27.95%) | 258.99 (28.87%) |
| Other/none | 1183 (21.29%) | 185 (36.56%) | | 1471.73 (22.3%) | 349.78 (38.99%) |
| Ethnicity | | | | | |
| White | 4498 (68.27%) | 412 (69.59%) | 0% | 5580.25 (84.55%) | 776.50 (86.55%)* |
| Mixed | 298 (4.52%) | 30 (5.07%) | | 165.52 (2.51%) | 23.70 (2.64%) |
| Indian | 439 (6.66%) | 15 (2.53%) | | 146.52 (2.22%) | 8.73 (0.97%) |
| Pakistani | 354 (5.37%) | 38 (6.42%) | | 158.68 (2.4%) | 21.96 (2.45%) |
| Bangladeshi | 289 (4.39%) | 41 (6.93%) | | 76.53 (1.16%) | 12.84 (1.43%) |
| Black African | 197 (2.99%) | 28 (4.73%) | | 95.29 (1.44%) | 26.65 (2.97%) |
| Black Caribbean | 278 (4.22%) | 12 (2.03%) | | 162.33 (2.46%) | 10.42 (1.16%) |
| Other | 236 (3.58%) | 16 (2.7%) | | 214.77 (3.25%) | 16.34 (1.82%) |
| Financial difficulties | | | | | |
| Managing well | 3407 (53.28%) | 203 (35.74%) | 4% | 3336.00 (51.37%) | 272.56 (31.38%)* |
| Getting by | 2622 (41%) | 309 (54.4%) | | 2791.23 (42.98%) | 517.32 (59.55%) |
| Having difficulties | 366 (5.72%) | 56 (9.86%) | | 367.05 (5.65%) | 78.83 (9.07%) |
| # Household children | | | | | |
| 1 | 1428 (22.03%) | 139 (24.13%) | 2.6% | 1494.96 (23.02%) | 237.04 (27.29%)* |
| 2 | 2811 (43.36%) | 210 (36.46%) | | 2857.47 (44%) | 312.58 (35.98%) |
| 3 | 1423 (21.95%) | 132 (22.92%) | | 1399.61 (21.55%) | 196.44 (22.61%) |
| 4 | 576 (8.88%) | 47 (8.16%) | | 557.35 (8.58%) | 66.61 (7.67%) |
| 5+ | 245 (3.78%) | 48 (8.33%) | | 184.88 (2.85%) | 56.04 (6.45%) |
| Household type | | | | | |
| Two parent | 5202 (80.12%) | 396 (68.39%) | 2.4% | 4821.42 (74.24%) | 514.38 (59.21%)* |
| Single parent | 1291 (19.88%) | 183 (31.61%) | | 1672.87 (25.76%) | 354.33 (40.79%) |

*P<0.05 from Meng and Rubin[50] likelihood ratio test.
GCSE, General Certificate of Secondary Education; GHQ, General Health Questionnaire; IMD, Index of Multiple Deprivation; NS-SEC, National Statistics Socio-Economic Classification; NVQ, National Vocational Qualification.

## Sensitivity analysis

The results of the 20 000 models, representing a random selection of all possible combinations of the covariates, are displayed in figure 4. Each line represents a single model run for each decile of GHQ-12 scores at age 25. Overlaid on the figure are the results of the main fully adjusted models, with and without further control for current economic status. The results are very consistent regardless of the set of control variables used.

The results of models included further adjustment for adolescent socioeconomic position and family background are displayed in online supplemental figure S2, with results from the main analysis (ie, column C, table 2) also shown for comparison. Again, results are very similar,

regardless of the control variables used. The results of the sensitivity analysis using different cut-offs to define youth unemployment are shown in online supplemental figure S3. The same qualitative pattern of stronger associations at higher quantiles is observed in each case, but associations are typically stronger when longer durations are used to define unemployment, though confidence intervals are overlapping.

## DISCUSSION

Using a quantile regression approach, we find that, on average, individuals who were continuously unemployed for 6+ months between ages 18 and 20 have poorer mental

**Table 2** Main regression results

| Quantile | (A) | (B) | (C) | (D) | (E) | (F) |
|---|---|---|---|---|---|---|
| Q10 | 0 (−1, 1) | 0 (−0.79, 0.57) | 0.15 (−0.67, 0.98) | 0.1 (−0.78, 0.95) | 0.83 (−0.26, 1.78) | −0.49 (−1.55, 0.87) |
| Q20 | 0 (0, 1) | 0.36 (−0.31, 0.93) | 0.49 (−0.2, 1.16) | 0.42 (−0.3, 1.03) | 0.79 (−0.13, 1.6) | 0.31 (−1.07, 1.35) |
| Q30 | 1 (0, 2) | 0.48 (−0.15, 1.2) | 0.51 (−0.19, 1.2) | 0.28 (−0.4, 0.94) | 0.61 (−0.32, 1.52) | 0.49 (−0.52, 1.43) |
| Q40 | 1 (0, 2) | 0.67 (−0.07, 1.57) | 0.55 (−0.16, 1.28) | 0.22 (−0.46, 0.92) | 0.59 (−0.35, 1.65) | 0.46 (−0.53, 1.46) |
| Q50 | 1 (0, 2) | 1.1 (0.35, 1.79) | 0.73 (−0.05, 1.54) | 0.22 (−0.59, 1.09) | 0.75 (−0.33, 1.92) | 0.58 (−0.64, 1.96) |
| Q60 | 2 (0, 2) | 1.57 (0.67, 2.48) | 1.1 (0.14, 2.32) | 0.52 (−0.4, 1.73) | 0.99 (−0.28, 2.51) | 1.19 (−0.32, 3.03) |
| Q70 | 3 (2, 4) | 2.41 (1.27, 3.85) | 2.03 (0.72, 3.36) | 1.55 (0.07, 2.82) | 1.76 (−0.07, 4.05) | 2.11 (0.18, 3.84) |
| Q80 | 4 (3, 6) | 4 (1.95, 6.2) | 2.95 (1.12, 4.92) | 2.24 (0.73, 4.02) | 3.22 (0.47, 6.4) | 2.3 (0.33, 4.36) |
| Q90 | 5 (3, 8) | 6.2 (3.22, 7.62) | 3.76 (1.82, 5.83) | 2.69 (0.92, 4.42) | 4.16 (1.21, 7.14) | 2.48 (0.34, 4.59) |
| Observations | 7363 | 7363 | 7363 | 7363 | 3196 | 4167 |
| Imputations | 28 | 28 | 28 | 28 | 28 | 28 |

health at age 25, but that there is substantial heterogeneity with associations being more pronounced at quantiles of the distribution representing the worse mental health. Effect sizes at poorer levels of mental health are substantial and clinically significant: a higher proportion of those who were unemployed as youths exceed suggested thresholds for detecting depression with GHQ-12 Likert scores.[32] Associations remain after adjusting for adolescent mental and physical health, suggesting results may not be explained by health-related selection into unemployment, though there appears to be measurement error in the instruments we use. Associations were not

fully attenuated when adjusting for current employment status and were robust to using different combinations of control variables. Point estimates were larger for males than females, particularly at higher quantiles of GHQ-12 scores.

Our results are consistent with previous studies which show that youth unemployment is related to worse mental health later in life[9–13 42] irrespective of later employment outcomes,[12] but we extend the literature by showing that associations differ substantially across individuals. A natural next question is to identify individuals who appear to be most impacted by youth unemployment. Associations in our sample were stronger among males

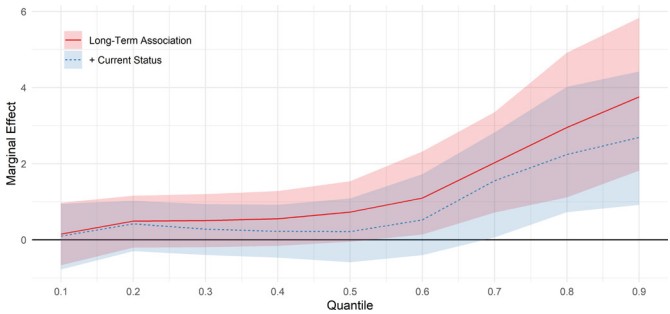

**Figure 1** Association between 6+ months youth unemployment between ages 18 and 20 and GHQ-12 Likert scores at age 25, by decile of GHQ-12. Solid line: adjusted for adolescent mental health, disability and self-rated health, educational attainment, risk behaviours, attitude to school and bullying victimisation, IMD quintile, parental socioeconomic class and education, gender, locus of control and ethnicity. Dashed line: additionally adjusted for current status. GHQ, General Health Questionnaire; IMD, Index of Multiple Deprivation.

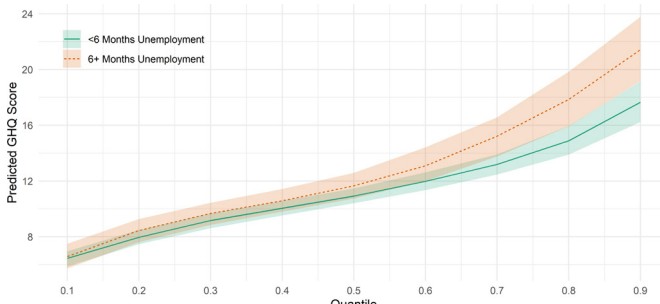

**Figure 2** Predicted age 25 GHQ-12 Likert scores by youth unemployment experience, by decile of GHQ-12. Models include adjustment for adolescent mental health, disability and self-rated health, educational attainment, risk behaviours, attitude to school and bullying victimisation, IMD quintile, parental socioeconomic class and education, gender, locus of control and ethnicity. Predicted values derived used sample means for covariates. GHQ, General Health Questionnaire; IMD, Index of Multiple Deprivation.

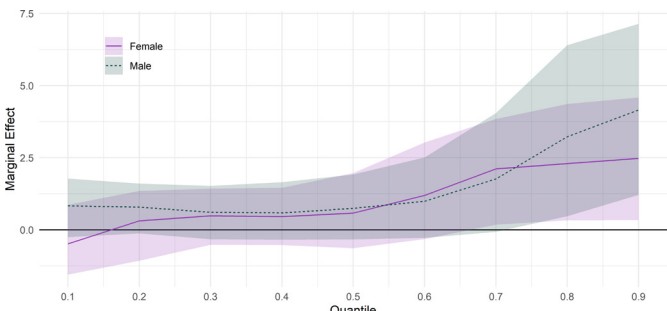

**Figure 3** Association between 6+ months youth unemployment between ages 18 and 20 and GHQ-12 Likert scores at age 25, by gender and decile of GHQ-12. Adjusted for adolescent mental health, disability and self-rated health, educational attainment, risk behaviours, attitude to school and bullying victimisation, IMD quintile, parental socioeconomic class and education, gender, locus of control and ethnicity. GHQ, General Health Questionnaire; IMD, Index of Multiple Deprivation.

than females (though CIs overlapped), which is consistent with the results of a recent Finnish study.[15] Men are generally found to experience worse mental health effects from (contemporary) unemployment,[2] though whether this applies to situations where similar numbers of males and females participate in the labour market has been questioned.[29] Further, studies also show differences in *economic* scarring effects by gender, which could also contribute to long-term negative mental health impacts, but the direction of this difference is not consistent across studies.[43 44]

Besides gender, a small number of moderating factors have been explored in the existing literature. Lee *et al*[9] find little evidence of differences by neighbourhood deprivation (though analyses appear underpowered), while Bijlsma *et al*[15] find evidence that scarring effects are stronger among males with low education, and Clark and Lepinteur[22] find evidence of stronger associations between early unemployment and later life satisfaction among males from disadvantaged households. Other possible moderators that deserve further exploration

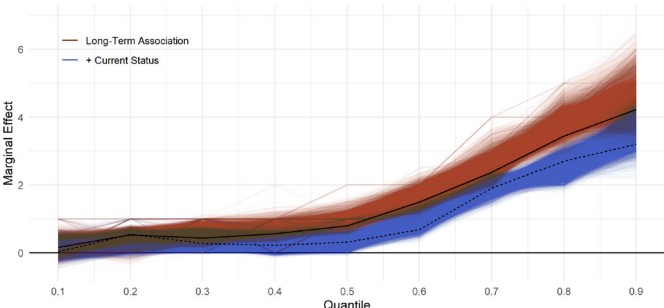

**Figure 4** Association between 6+ months youth unemployment between ages 18 and 20 and GHQ-12 Likert scores at age 25, by decile of GHQ-12. A total of 20 000 models drawn from all possible combinations of covariates. Black lines are results of fully adjusted models, including (dashed) or not including (solid) further adjustment for current economic activity. GHQ, General Health Questionnaire.

are personality traits, such as locus of control. Locus of control has been linked to psychological adjustment following adversity[45 46] and also to active job-seeking behaviours among the unemployed.[47] Identifying vulnerable groups would have important implications for the targeting of policy interventions.

Another avenue for future research is to identify the mechanisms through which scarring effects may operate. Bijlsma *et al*[15] find evidence that the association between unemployment and later depression is partly mediated by income, but we know of no study that has assessed whether unemployment during adolescence alters neurobehavioural development. These pathways have different implications for designing policy to reduce scarring effects and potentially for identifying vulnerable groups. At present we are unable to say whether the heterogeneity observed reflects individual differences in psychological resilience or propensities for unemployment to engender future socioeconomic adversity.

Our study has a number of limitations. We use observational data and as such, our results cannot be taken as indicating causality. Associations may reflect health-related selection into unemployment, though we adjust for adolescent mental and physical health and differences in GHQ scores were observed across most of the distribution—mental health-related selection into unemployment is arguably only a plausible explanation where psychological morbidity is severe enough to preclude job-finding or job-holding. Nevertheless, there are several other factors that are unobserved or imperfectly measured in our study, such as personality traits and human capital, which may explain associations. Controlling for current economic activity is likely to have induced collider bias. Estimates may have appeared more attenuated than was accurate. While we included attrition weights in our analysis, over half of the sample did not participate at the age 25 survey. If participants who were most harmed by unemployment were more likely to drop-out of the survey, this would also bias results.[43]

Another limitation is that, by focusing on unemployment during ages 18–20, we exclude from the unemployed group those that only enter the labour market at a later age (ie, those that went to university). Previously research suggests scarring effects are smaller among this group.[15] Our use of the GHQ as a measure of mental health could also have biased results as items inquire about current symptoms vis-à-vis typical experience. This may not appropriately capture chronic psychological ill health, though a recent validation study finds that the GHQ displays good sensitivity for detecting depression.[32]

Our results may be specific to the cohort we studied. The period in which we measure youth unemployment overlapped with the 2008 Global Financial Crisis, following which youth unemployment rates rose worldwide.[1] The long-term consequences of unemployment may be smaller following a recession as prospective employers may look on periods of unemployment less unfavourably,[48] and so social chains of risk may be weaker.

Further, outcomes for those that do not become unemployed—who we use to measure long-term effects—may simultaneously get worse as labour markets get weaker.[49]

Finally, the results may be specific to the age we studied. Participants were early in their labour market careers when we measured mental health outcomes. Differences between the youth unemployed and their peers may grow through time (eg, following promotions) or become progressively more important if, for instance, individuals are less able to rely on financial help from their families as they age. Alternatively, differences could dissipate if individuals are able to recover from early labour market adversity and it is 'chains of risk' that explain our results. A useful extension to this study would be to repeat the analysis at later ages and in other countries with different labour market institutions.

**Contributors** All authors contributed to the design of the study. LW carried out the analysis and wrote the manuscript. JH and SJ provided detailed comments on the manuscript.

**Funding** This work was funded by the Economic and Social Research Council through the UCL, Bloomsbury and East London Doctoral Training Partnership (ES/P000592/1). The funding was awarded to LW.

**Competing interests** None declared.

**Patient and public involvement** Patients and/or the public were not involved in the design, or conduct, or reporting, or dissemination plans of this research.

**Patient consent for publication** Not required.

**Ethics approval** Ethical approval for Next Steps has been gained from the National Health Service (NHS) Research Ethics Committee (REC) system (14/LO/0096). This is a secondary data analysis of Next Steps so ethical approval was not required.

**Provenance and peer review** Not commissioned; externally peer reviewed.

**Data availability statement** Data are available in a public, open access repository. The code to replicate this analysis is available at https://osf.io/3gwap/. Next Steps data are available through the UK Data Service (http://doi.org/10.5255/UKDA-SN-5545-7).

**ORCID iD**
Liam Wright http://orcid.org/0000-0002-6347-5121

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
