## [Reviewer comments · BMJ Open]

ARTICLE DETAILS

TITLE (PROVISIONAL)	Heterogeneity in the Association Between Youth Unemployment and Mental Health Later in Life: A Quantile Regression Analysis of Longitudinal Data from English Schoolchildren
AUTHORS	Wright, Liam; Head, Jenny; Jivraj, Stephen

VERSION 1 – REVIEW

REVIEWER	Mauro , Joseph University of Central Arkansas
REVIEW RETURNED	26-Feb-2021

GENERAL COMMENTS	Summary This study uses a longitudinal study to exam how extended durations of unemployment during young adult years may impact mental health. The authors implement a quantile regression strategy to explore any potential association. They find that youth unemployment is related to worse mental health at age 25 but with a great deal of heterogeneity across quantiles. Overall, while there are some limitations to the study it helps further the body of research examining the link between youth unemployment and mental health outcomes. Comments • Page 2, Bullet #4: typo – “test whether are results are” should be “test whether our results are”• There is a large body of work that consistently finds that youth workers are especially more susceptible to unemployment during financial crises – (Islam & Verick 2011). In particular male workers are more at risk of unemployment. Overall, despite the authors mentioning this there is little discussion of outcomes across gender. While the results are in Table 2 (Columns E & F) there is little analysis as to how they are different. Authors should elaborate here as it appears that in the male only model the predicted GHQ score is higher for males than females. This may be because male workers are more at risk than
--

	female workers as some studies such as Islam & Verick have found.  • It may be of interest to interact unemployment with other covariates such as gender and educational levels to gain more insight into these results and help identify which groups are most at risk to the mental health issues stemming from youth unemployment. • Page 4, Line 40 – “15,7770” is a typo • From the description of the sample, it seems that a high proportion of individuals included were from schools with free meals. This might introduce some bias as these individuals may be coming from families of lower socioeconomic backgrounds and as such may be more/less susceptible to mental health issues. This should be discussed further. • Covariates – Do the authors have any other variables to control for household environment (i.e., female head of household, number of children in household, etc.)? Again, this may help to better identify the population being examined. • Page 6 – Youth Unemployment: How is Youth Unemployment being defined? Are these individuals continuously looking for jobs or are the marginally attached in that they want to work but are discouraged? • Are individuals with no unemployment experience during the sample period included in the <6+ month group? If so, have the authors tried altering their framework to either look solely at those who have bouts of unemployment less than 6+ months (not including those who have never experienced unemployment) and those that have 6+ months. This may strengthen results if significance is maintained and highlight long-term unemployment as a key to story. • Page 8 Line 59 – Table Reference # Error • While the authors touch upon this, there is limited discussion about youth idleness or inactiveness. The youth unemployment rate is not always the best measure of labor market performance as it fails to capture youths who involved with acquiring more education or training. As such it is sometimes better to examine youths who are not in education or training (sometimes called NEETs). This is a limitation of the study that should be discussed in more detail. • While it is clear that that focus on this study is on long-term unemployment at 6+ months have the authors examined the sensitivity to the 6+ month mark. For instance, can we determine at what month threshold youth unemployment impacts mental health outcomes. It may be of interest to replicate these results for 3+, 9+ or 12+ months to see how results at different quantiles changes.
--	---

REVIEWER	Dwyer, Rocky J
REVIEW RETURNED	Walden University, College of Management and Technology 07-Apr-2021

GENERAL COMMENTS	I would recommend that the authors have a profession edit since there are some minor errors in flow and readability related to tense.
---

VERSION 1 – AUTHOR RESPONSE

Reviewer 1

This study uses a longitudinal study to exam how extended durations of unemployment during young adult years may impact mental health. The authors implement a quantile regression strategy to explore any potential association. They find that youth unemployment is related to worse mental health at age 25 but with a great deal of heterogeneity across quantiles. Overall, while there are some limitations to the study it helps further the body of research examining the link between youth unemployment and mental health outcomes.

Comment	Response
Page 2, Bullet #4: typo – “test whether are results are” should be “test whether our results are”	Thank you for reviewing our work. We have amended this error.
There is a large body of work that consistently finds that youth workers are especially more susceptible to unemployment during financial crises – (Islam & Verick 2011). In particular male workers are more at risk of unemployment. Overall, despite the authors mentioning this there is little discussion of outcomes across gender. While the results are in Table 2 (Columns E & F) there is little analysis as to how they are different. Authors should elaborate here as it appears that in the male only model the predicted GHQ score is higher for males than females. This may be because male workers are more at risk than female workers as some studies such as Islam & Verick have found.	We did not discuss this result much given the overlapping confidence intervals in the results by gender. An issue is that youth unemployment is relatively uncommon, so interaction terms or cross-group comparisons would be underpowered. Nevertheless, we agree that it is still interesting to discuss the result, so we have now included further text on this in the Discussion: “Associations in our sample were stronger among males than females (though confidence intervals overlapped), which is consistent with the results of a recent Finnish study [15]. Men are generally found to experience worse mental health effects from (contemporary) unemployment [2], though whether this applies to situations where similar numbers of males and females participate in the labour market has been questioned [29]. Further, studies also show differences in economic scarring effects by gender, which could also contribute to long-term negative mental health impacts, but the direction of this difference is not consistent across studies [43,44].”
It may be of interest to interact unemployment with other covariates such as gender and educational levels to gain more insight into these results and help identify which groups are most at risk to the mental health issues stemming from youth unemployment.	We agree that this would be an important insight. However, as noted above, we do not have statistical power to conduct such an analysis, given that youth unemployment is relatively uncommon in our sample. Therefore, we have not run these extra analyses, but we have amended our discussion of papers that have attempted to answer this question. Specifically, the Discussion now reads: “Besides gender, a small number of moderating factors

	have been explored in the existing literature. Lee et al. [9] find little evidence of differences by neighbourhood deprivation (though analyses appear underpowered), while Bijlsma and colleagues [15] find evidence that scarring effects are stronger among males with low education, and Clark and Lepinteur [22] find evidence of stronger associations between early unemployment and life satisfaction among males from disadvantaged households.”
Page 4, Line 40 – “15,7770” is a typo	Thank you for spotting this. We have now amended this.
From the description of the sample, it seems that a high proportion of individuals included were from schools with free meals. This might introduce some bias as these individuals may be coming from families of lower socioeconomic backgrounds and as such may be more/less susceptible to mental health issues. This should be discussed further.	We have included attrition weights in our analysis that account for the design of the survey, so these individuals should not be overrepresented in the final results. However, over half of the sample had dropped out by age 25, so we have now included the possibility of attrition bias as a limitation of the analysis: “While we included attrition weights in our analysis, over half of the sample did not participate at the age 25 survey. If participants who were most harmed by unemployment were more likely to drop-out of the survey, this would also bias results [though, see, 43].”
Covariates – Do the authors have any other variables to control for household environment (i.e., female head of household, number of children in household, etc.)? Again, this may help to better identify the population being examined.	Next Steps is a very detailed survey containing a lot of information on the adolescent household environment. While we considered including more covariates in our models, we selected variables considering the pathways through which socio-economic background is likely to influence youth unemployment and mental health (VanderWeele, 2019). Nevertheless, as a further sensitivity analysis, we have now run models including further adjustment for: number of household children, number of parents in household, and household financial difficulties (Supplementary Figure S2). The results are almost identical to those presented in the main analysis. In the amended analysis, we included these variables in imputation models, which means results are now slightly different (numerically, but not qualitatively) from the previous submission. (Note, we now also use sample means and modes when calculating predicted values [Figure 2], so confidence intervals in the figure are wider but results are qualitatively the same.)
Page 6 – Youth Unemployment: How is Youth Unemployment being defined? Are these individuals continuously looking for jobs or are	At each wave, participants were asked for the set of main activities they had carried out since the previous wave. Activities were chosen from

the marginally attached in that they want to work but are discouraged?	a list. Unemployment appeared as “Unemployed” or “Unemployed and looking for work”, depending on the wave. There were also categories for various economically inactive statuses, such as homemaking, travelling, or taking a break from education. We did not use these to define unemployment. We have added further detail to the Measures subsection to clarify the measurement of unemployment.
Are individuals with no unemployment experience during the sample period included in the <6+ month group? If so, have the authors tried altering their framework to either look solely at those who have bouts of unemployment less than 6+ months (not including those who have never experienced unemployment) and those that have 6+ months. This may strengthen results if significance is maintained and highlight long-term unemployment as a key to story.	The comparator group is all individuals with < 6 months unemployment, including those with no unemployment, whatsoever. Unfortunately, there are only 225 individuals who have 1-5 months unemployment, so we do not have statistical power to run this analysis.
Page 8 Line 59 – Table Reference # Error	Thank you for spotting this. We have now fixed this.
While the authors touch upon this, there is limited discussion about youth idleness or inactiveness. The youth unemployment rate is not always the best measure of labor market performance as it fails to capture youths who involved with acquiring more education or training. As such it is sometimes better to examine youths who are not in education or training (sometimes called NEETs). This is a limitation of the study that should be discussed in more detail.	While we agree that NEET is important, we focused on youth unemployment because it captures a more clearly defined, homogeneous set of experiences. As Yates and Payne (2006) note, NEET includes full-time carers, people who are voluntarily NEET (e.g. those on gap years) and those facing more salient risks than non-participation in work, education or training, such as homelessness or chronic health problems. Though there is a small literature showing NEET individuals have worse long-term outcomes than their peers (see, for example, Ralston et al., 2016), it is unclear which NEET subgroups results apply to and unlikely they will apply to all. As Furlong (2006, p. 555) notes, the heterogeneity within NEET “means that both research and policy must begin by disaggregating so as to be able to identify the distinct characteristics and needs of the various sub-groups”. Another reason for focusing on youth unemployment is that the literature on the long-term consequences of youth unemployment is much larger than that for the consequences of NEET. As our results adds a caveat to the existing results (average differences mask substantial heterogeneity), it is important that we focus on youth unemployment. We have now included an explanation for why we have focused on youth unemployment in the Measures section.

While it is clear that that focus on this study is on long-term unemployment at 6+ months have the authors examined the sensitivity to the 6+ month mark. For instance, can we determine at what month threshold youth unemployment impacts mental health outcomes. It may be of interest to replicate these results for 3+, 9+ or 12+ months to see how results at different quantiles changes.	We have now included a further sensitivity analysis comparing 3+, 6+, 9+, and 12+ months cut-offs to define youth unemployment. Unfortunately, Next Steps is not adequately powered to test statistically for differences in the coefficients using these different cut-offs, but the results are as expected: the association is strongest if larger cut-offs are used (Supplementary Figure S3).
--	--

Reviewer 2

Comment	Response
I would recommend that the authors have a profession edit since there are some minor errors in flow and readability related to tense.	Thank you for reviewing our work. Each of the authors has re-proofread the manuscript.

References

- Furlong, A. (2006). Not a very NEET solution. *Work, Employment and Society*, 20(3), 553–569. <https://doi.org/10.1177/0950017006067001>
- Ralston, K., Feng, Z., Everington, D., & Dibben, C. (2016). Do young people not in education, employment or training experience long-term occupational scarring? A longitudinal analysis over 20 years of follow-up. *Contemporary Social Science*, 11(2–3), 203–221. <https://doi.org/10.1080/21582041.2016.1194452>
- VanderWeele, T. J. (2019). Principles of confounder selection. *European Journal of Epidemiology*, 34(3), 211–219. <https://doi.org/10.1007/s10654-019-00494-6>
- Yates, S., & Payne, M. (2006). Not so NEET? A critique of the use of ‘NEET’ in setting targets for interventions with young people. *Journal of Youth Studies*, 9(3), 329–344. <https://doi.org/10.1080/13676260600805671>

VERSION 2 – REVIEW

REVIEWER	Mauro , Joseph University of Central Arkansas
REVIEW RETURNED	05-Jul-2021

GENERAL COMMENTS	None needed. I felt all of my concerns were addressed from the previous revision.
---

REVIEWER	Dwyer, Rocky J Walden University, College of Management and Technology
REVIEW RETURNED	19-Jun-2021

GENERAL COMMENTS

Thank you for revising your paper to ensure flow and readability is consistent throughout the paper. I look forward to further research you conduct, and I have every confidence you will gain reader interest and a following for your research.